# Ultrasensitive Diamond Microelectrode Application in the Detection of Ca^2+^ Transport by AnnexinA5-Containing Nanostructured Liposomes

**DOI:** 10.3390/bios12070525

**Published:** 2022-07-14

**Authors:** Alberto Pasquarelli, Luiz Henrique Silva Andrilli, Maytê Bolean, Claudio Reis Ferreira, Marcos Antônio Eufrásio Cruz, Flavia Amadeu de Oliveira, Ana Paula Ramos, José Luis Millán, Massimo Bottini, Pietro Ciancaglini

**Affiliations:** 1Institute of Electron Devices and Circuits, University of Ulm, 89069 Ulm, Germany; 2Department of Chemistry, Faculty of Philosophy, Sciences and Letters at Ribeirão Preto, University of São Paulo, Ribeirão Preto 14040-901, SP, Brazil; luiz.andrilli@usp.br (L.H.S.A.); maytebolean@usp.br (M.B.); claudio.dos_reis_ferreira@sorbonne-universite.fr (C.R.F.); marcos.antonio.cruz@usp.br (M.A.E.C.); anapr@ffclrp.usp.br (A.P.R.); 3Sanford Burnham Prebys, La Jolla, CA 92037, USA; famadeu@sbpdiscovery.org (F.A.d.O.); millan@sbpdiscovery.org (J.L.M.); 4Department of Experimental Medicine, University of Rome Tor Vergata, 00133 Rome, Italy; massimo.bottini@uniroma2.it

**Keywords:** potential transient, diamond microelectrode, Annexin A5, proteoliposome, selective calcium transport

## Abstract

This report describes the innovative application of high sensitivity Boron-doped nanocrystalline diamond microelectrodes for tracking small changes in Ca^2+^ concentration due to binding to Annexin-A5 inserted into the lipid bilayer of liposomes (proteoliposomes), which could not be assessed using common Ca^2+^ selective electrodes. Dispensing proteoliposomes to an electrolyte containing 1 mM Ca^2+^ resulted in a potential jump that decreased with time, reaching the baseline level after ~300 s, suggesting that Ca^2+^ ions were incorporated into the vesicle compartment and were no longer detected by the microelectrode. This behavior was not observed when liposomes (vesicles without AnxA5) were dispensed in the presence of Ca^2+^. The ion transport appears Ca^2+^-selective, since dispensing proteoliposomes in the presence of Mg^2+^ did not result in potential drop. The experimental conditions were adjusted to ensure an excess of Ca^2+^, thus confirming that the potential reduction was not only due to the binding of Ca^2+^ to AnxA5 but to the transfer of ions to the lumen of the proteoliposomes. Ca^2+^ uptake stopped immediately after the addition of EDTA. Therefore, our data provide evidence of selective Ca^2+^ transport into the proteoliposomes and support the possible function of AnxA5 as a hydrophilic pore once incorporated into lipid membrane, mediating the mineralization initiation process occurring in matrix vesicles.

## 1. Introduction

Nanocrystalline diamond (NCD) has proven to be an excellent material for the fabrication of microelectrodes and microelectrodes-arrays (MEAs). Although an electrical insulator, diamond can reach quasi-metallic conductivity when doped with boron. Boron-doped nanocrystalline diamond (BNCD) microelectrodes present excellent properties such as very low background current (<1 μA/cm^2^), wide water dissociation potential window (~3 V), high biocompatibility and inertness in harsh environments. In addition, BNCD microelectrodes are suitable for measuring electrochemical potentials, showing sensitivity in good agreement with Nernst’s law [1,2].

Several applications can be found for BNCD electrodes and microelectrodes, including treatment of polluted waters [3,4], electroactive implants [5,6,7], electroanalysis [8], drugs analysis [9], and electrophysiology [10]. In the field of life sciences, they have been applied most often for the amperometric electrochemical detection of secretory events (exocytosis) from neuroendocrine cells [11,12,13,14,15] and for measuring action potentials [16,17,18], both in vitro and in vivo [5]. Indeed, the changes in the concentration of divalent ions such as Ca^2+^ and Mg^2+^ can be determined with the aid of BNCD microelectrodes, which allow their application for the investigation of ion-dependent biological processes [1,2].

Annexin A5 (AnxA5) is responsible for many physiological events such as the inhibition of phospholipid-dependent blood clotting [19,20], modulation of protein kinase C [21], inhibition of phospholipase A2 activity [22], formation of Ca^2+^ channels [22,23,24,25,26] and high affinity to type-II collagen, as described by our group using AnxA5-containing proteoliposomes [27]. In addition, the participation of AnxA5 in cellular disorganization resulting in apoptosis processes has been described [28,29]. A less exploited property of proteins belonging to the annexin family is the ability to induce curvature in lipid membranes, which can be of crucial importance for their functions as well as for their interaction with the components of lipid bilayer membranes [29,30,31,32].

Human AnxA5 (~ 35 kDa) was the first to have its three-dimensional structure characterized [33,34]. It is a slightly curved molecule, consisting of four homologous repetitions of about 70 amino acids, forming compact domains of five α-helices [34]. These domains are arranged forming a hydrophilic pore in the center of the protein that serves as a Ca^2+^ channel specifically in the membrane of matrix vesicles (MVs) [28,29,35,36,37]. The addition of specific anti-annexin antibody or the annexin channel inhibitor K-201 [38,39,40] decreases Ca^2+^ concentration, thus hampering mineralization by MVs [41,42]. Moreover, the addition of exogenous AnxA2, AnxA5 and AnxA6 to non-mineralizing annexin-deficient MVs restored their Ca^2+^ accumulation ability [41]. To understand how annexins can mediate the influx of Ca^2+^ in MVs, all factors that affect the function of these proteins during the mineralization process must be identified, such as their locations and their specific functions within the mineralization process as a whole.

In this paper, we describe the use of a BNCD microelectrode to track Ca^2+^ transport mediated by AnxA5 when inserted into the lipid bilayer of liposomes (proteoliposome) as MV-mimetic systems. Since the measurements technique using a BNCD microelectrode proved to be highly sensitive, we were able to evaluate the selectivity of the transport mechanism.

## 2. Materials and Methods

### 2.1. Chemicals and Solutions

All aqueous solutions were prepared with ultrapure apyrogenic water from a Millipore DirectQ^®^ system. Analytical grade bovine serum albumin (BSA), tris-hydroxymethyl- amino-methane (Tris), sodium dodecylsulfate (SDS), polyoxyethylene-9-lauryl ether (polidocanol), 1,2-dipalmitoyl-sn-glycero-3-phosphocholine (DPPC), 1,2-dipalmitoyl- snglycero-3-phospho-L-serine (DPPS), calcium chloride, magnesium chloride, sodium chloride, chloroform and ethylenediaminetetraacetic acid tetrasodium (EDTA) were purchased from Merck KGaA, St. Louis, MO, USA and used without further purification.

### 2.2. Preparation of Liposomes and Proteoliposomes

Liposomes constituted by 9:1 DPPC:DPPS (molar ratio), containing up to 1.5 mg/mL of lipids were prepared as previously described [27]. Briefly, lipids were dissolved in chloroform and dried under nitrogen flow. The resulting lipid film was kept under vacuum overnight and suspended in the stock buffer (16 mM Tris-HCl buffer, pH 7.5), to obtain a final lipid solution of 1.5 mg/mL. The mixture was then incubated for 1 h at 70 °C, above the critical phase transition temperature of the lipid, and vortexed at 10 min intervals. Large unilamellar vesicles (LUVs) were prepared by submitting the suspension to extrusion through 100 nm polycarbonate membranes in a LiposoFast extrusion system (Merck KGaA, St. Louis, MO, USA). AnxA5 (35 kDa), expressed in E. coli as described by Bolean et al. [26], was incorporated into 9:1 DPPC:DPPS-liposomes (molar ratio) by direct insertion in the stock buffer, resulting in a 1:100 (protein:lipid molar ratio). Then, the mixture was ultracentrifuged at 100,000× *g* for 1 h at 4 °C. The pellet containing proteoliposomes was resuspended in an appropriate volume of the same buffer resulting in a 0.6 mg/mL of total lipid. Liposomes and proteoliposomes were characterized by dynamic light scattering (DLS) and zeta (ζ)-potential measurements. Protein concentration incorporated in the lipid bilayer of liposomes was estimated in the presence of 2% (*w/v*) SDS, and BSA was used as standard [43].

### 2.3. Atomic Force Microscopy (AFM) Analyses of Liposome and Proteoliposome

Liposome and proteoliposome samples were prepared with stock buffer and stabilized by adding 1:1 (*v/v*) glutaraldehyde (~5% final concentration) to avoid vesicle deformation and disruption. The mixtures were homogenized and then 5 μL of the sample was dropped onto freshly cleaved mica substrate, left to dry at room temperature and imaged by AFM as described by Bolean et al. [44]. AFM micrographs were obtained by a Shimadzu SPM-9600 Scanning Probe Microscopy (Shimadzu Corporation, Kyoto, Japan) operating in tapping mode (called Phase Contrast Mode by Shimadzu), which is a dynamic based mode. Scanning was performed in air at 25 °C using standard four-sided pyramidal silicon probes with a resonance frequency ranging from 324 to 369 kHz (Nanosensors™, Neuchatel, Switzerland). The scan rate was set at 0.2–0.3 Hz to prevent tip-induced vesicle deformations and/or damage. The diameters of vesicles and protrusions were determined by SPM Offline software from Shimadzu. For each sample, *n* = 100 vesicles were analyzed.

### 2.4. Potential Measurements Using Diamond Microelectrodes

A planar four-channel BNCD microelectrodes array was fabricated as previously described in detail by Gao et al. [45], with a slightly different layout for providing four individual microelectrode openings in the passivation. Briefly, the process consisted of the following steps:

(1) Clean a blank sapphire wafer.

(2) Deposit a ~60 nm layer of amorphous silicon (α-Si) for buffering the thermal expansion mismatch between sapphire and diamond and promote the diamond nucleation.

(3) Grow a ~500 nm thick film of intrinsic NCD.

(4) Grow a ~250 nm film of BNCD.

(5) Transfer the pattern of the conducting structures via optical lithography, metal mask protection and reactive ion etching (RIE) in an argon-oxygen atmosphere.

(6) Deposit a 1 µm thick passivation layer of silicon nitride by plasma-enhanced CVD (PECVD).

(7) Transfer the pattern of the microelectrode openings (16 µm in diameter) and bonding pads by optical lithography and RIE in a tetrafluoromethane (CF_4_) atmosphere.

(8) Dice the wafer for obtaining the individual chips.

After fabrication, the chip was assembled on a printed-circuit carrier plate by flip-chip bonding with conducting silver-epoxy, and finally a glass ring was glued over the device for providing a 200 µL volume for the biological environment (Figure 1). A ring-shaped Ag/AgCl quasi-reference electrode was immersed in this volume and connected to the ground potential. For the in-house fabricated readout electronics, we adopted a voltage follower design with 1 pA input bias current and 10 TΩ input impedance. This unitary gain stage was followed by a Bessel low-pass filter of the 4th order with a cut-off frequency of 1 kHz. All four conditioned signals were acquired over differential inputs with a USB-6216 data acquisition unit (National Instruments, Austin, TX, USA) at a sampling rate of 4 kHz per channel and a resolution of 16 bit. For running the experiments, we designed a LabVIEW application including a linear phase digital filter with decimation, which delivered output signals sampled at 80 Hz over a bandwidth of 36 Hz.

Before the first use, the device was exposed to mild oxygen plasma for 10 min to provide the desired surface termination. This step is of fundamental importance for potentiometric measurements. In fact, oxygen termination is not only responsible for a hydrophilic behavior, but results in a surface band bending and a stable pinning of the surface Fermi level at 1.7 eV above the valence band, thus making it reliably sensitive to variations of the electrochemical potential in the electrolyte [46].

In the adopted configuration, the chip is not provided with a calcium-specific ion-selective membrane. This is intentional, in order to measure with the same device, the response of liposomes and proteoliposomes to various ions, thus assessing their selectivity and specificity. Therefore, the microelectrodes sense the cumulative electrochemical potential resulting from all ions present in the electrolyte. This potential *E* can be estimated by means of the general Nernst equation by taking into account the activity of all oxidized and reduced species $a_{i,ox}$ and $a_{j,red}$ with their respective stoichiometric coefficients $\nu_{i}$ and $\nu_{j}$ and charge z as shown below:(1)$$E=E^{Ө}+\frac{RT}{zF}ln\left(\frac{\prod_{i}\left[a_{i,ox}^{\nu_{i}}\right]}{\prod_{j}\left[a_{j,red}^{\nu_{j}}\right]}\right)$$
with *E*^Ө^ the potential offset given by the Ag/AgCl quasi-reference electrode, *R* the universal gas constant, *F* the Faraday constant and *T* the absolute temperature. Potential changes are due to concentration changes of certain species, including buffering effects. This means that the contribution of calcium ions to the measured potential is only one among many, and the presence of a buffer can compensate, at least partially, the potential variations due to ion-transport inside the vesicles.

After every measurement session, the device was cleaned with a 14% solution of sodium hypochlorite to remove organics and restore a conveniently oxidized surface, then abundantly rinsed with Millipore DirectQ ultra-pure apyrogenic water, and finally dried with a nitrogen gun. No material degradation could be observed by optical inspection, even after several measuring/cleaning cycles. However, we observed from measurement to measurement slightly different baseline potentials after equilibration, which may indicate different characteristics of the surface termination. In fact, it is well known that oxygen can form ketone-bonds, ester bonds or hydroxyl groups when terminating a carbon surface. The random proportions of these three O-termination variants leads to slightly different chemical activity and surface potentials. During the measurements, the front-end electronics was held in a ferromagnetic shielding box, equivalent to a Faraday cage, having a small opening above the device to allow manual pipetting operations. Unfortunately, this simple technique introduced artifacts in the signal recordings, but these did not affect the long-term trends (300 s). Due to the uniformity of the biological sample, the signals detected from the four channels were substantially identical; therefore, only one channel was considered for data analysis. Data presented in the figures are representative of triplicate measurements using different vesicle preparations.

## 3. Results

### 3.1. Characterization of Liposomes and Proteoliposomes

The mean diameter and ζ-potential of 9:1 DPPC:DPPS-liposomes (molar ratio) were 110 ± 10 nm and −2.3 ± 0.1 mV, respectively. Proteoliposomes containing 14 µg/mL of AnxA5 had a diameter of 140 ± 15 nm and a ζ-potential −21.8 ± 0.3 mV. These vesicles were also topologically characterized by AFM and tested in vitro as biomimetics of the MV functions [26,27,44]. AFM is a versatile tool to perform morphologic analysis of liposomes and proteoliposomes at the nanoscale without the use of fluorescent dyes, enabling acquisition of the molecular details of the vesicle’s surface [44]. Figure 2 shows phase images of liposomes and proteoliposomes harboring AnxA5, both composed of 9:1 DPPC:DPPS (molar ratio).

DPPC:DPPS liposomes show spherical and intact (unbroken) vesicles. The presence of negative charges in the DPPS lipid results in a rough surface (Figure 2A), but no protrusion is observed (Figure 2B). Insertion of AnxA5 into DPPC:DPPS 9:1 (molar ratio) liposomes led to the formation of surface protrusions that are darker domains than the surrounding surface (Figure 2C). The protrusions formed by AnxA5 on the surface of liposomes displayed a height <0.5 nm, as previously observed [44]. Zoom of the select area showing the surface of one proteoliposome enabled observation of these protrusions in detail (Figure 2D). They are darker spots on the surface of the proteoliposomes, which are domains with lower viscoelasticity, suggesting the existence of regions on the surface of these proteoliposomes with different chemical compositions induced by the protein presence [44].

### 3.2. Potentiometric Measurements

The ability of AnxA5 in selectively binding and accumulating Ca^2+^ inside proteoliposomes was followed by using the BNCD microelectrodes described above, standardized and calibrated using the same method described by Denisenko et al. [2]. With the aid of this microelectrode array, changes in the potential generated by free Ca^2+^ in solution were measured. The experiment depicted in Figure 3 was run without Ca^2+^ ions. The potential of the microelectrode equilibrated after approximately 100 s, thus reaching a nearly flat baseline. The addition of Tris-HCl buffered liposomes or proteoliposomes into the microelectrode compartment (Figure 1f) after 300 s resulted in positive steps of ~150 and ~130 mV, respectively, in the recorded potential, which remained nearly constant as a function of time. Only a small potential drop was observed, indicating a second equilibration transient. Qualitatively similar behavior was detected with samples dispersed in either deionized water or 16 mM Tris-HCl buffer, pH 7.5, even in the presence of 2.5 mM NaCl (data not shown).

The test displayed in Figure 4 shows that the ion transport through the pore is apparently selective to Ca^2+^, since the addition of proteoliposomes did not result in a significant reduction of the potential with time in the presence of Mg^2+^ ions instead of Ca^2+^ ions (Figure 4, continuous red line). The initial rate of Ca^2+^ ion uptake resulted in a reduction of 2.5 mV/s in the potential, while for Mg^2+^ ions the rate was 0.3 mV/s (Figure 4, inset).

The potential increased and remained constant with time when liposomes were added to the microelectrode in the presence of 1 mM Ca^2+^ (Figure 4, dotted black line). The addition of proteoliposome at 300 s in the presence of 1 mM Ca^2+^ ions resulted in a potential step (Figure 4, continuous black line) similar to that observed without Ca^2+^ ions (Figure 3, continuous line). However, the potential values decreased very significantly with time, indicating an event different from the simple second equilibration (Figure 4, continuous black line). At 600 s, the potential was lower than the baseline, suggesting that Ca^2+^ ions were incorporated into the vesicle compartment (Figure 4, inset) and were no longer being detected by the microelectrode.

Under the conditions of the measurements carried out using the BNCD microelectrodes, the molar ratio of Ca^2+^/AnxA5 was 50. Since it has been reported that the AnxA5 monomer binds up to 10 Ca^2+^ ions [31], this experimental condition guaranteed excess of Ca^2+^ in the solution and confirmed that the reduction in the potential was not only due to the binding of Ca^2+^ to AnxA5 but mainly to the transfer of ions to the lumen of the proteoliposomes, showing the transport function of AnxA5 when incorporated in the vesicles’ membrane bilayer.

The addition of up to 2.5 mM of EDTA at time 240 s to the microelectrode chamber, filled from the beginning with Tris-HCl buffer, resulted in a fast decrease of the potential. This behavior is probably caused by the reduction of Ca^2+^ free ions concentration in the solution due to the formation of EDTA-Ca^2+^ complexes. In this way, the potential change recorded after dispensing proteoliposomes at time 300 s to the microelectrode in the presence of EDTA (Figure 5) is similar to the signal observed in the absence of ions (Figure 3). In other words, EDTA and proteoliposomes show similar effects regarding the reduction of the potential by the uptake of Ca^2+^ (Figure 5), differing only in the velocity of response, which appears quite consistent with the fast action of a chelator compared with the slow ion-transport in a membrane protein.

For further characterizing the individual behaviors of EDTA and proteoliposomes, in a next experiment the dispensing order was inverted. The measurements displayed in Figure 6 revealed that the potential reduction, due to the Ca^2+^ ions uptake in the proteoliposomes, stopped immediately after the addition of EDTA despite the step observed after dispensing the chelator. These data corroborate the hypothesis that the changes in the potential are related to the concentration of Ca^2+^ ions in the solution. Since these ions are chelated, the potential is stabilized.

A few more measurement variations were carried out for obtaining a better overview, including the influence of the dispensing sequence, the buffering effects and the selectivity of AnxA5 for Ca^2+^. Table 1 summarizes all recordings and their results.

## 4. Discussion

As recently described by Lin et al. [31], Mularski et al. [32] and Bolean et al. [26], despite being a soluble protein, AnxA5 has also amphitropic characteristics that drive its peripherally binding to the surface of the lipid membrane by means of a self-assembly process that results in the formation of protein-rich domains. These domains are arranged at the lipid bilayer to define a hydrophilic pore through the center of the protein. This process results in the modulation of the physical properties of the membrane, such as thickness and diffusivity, indicative of a phase transition from the fluid to a more ordered phase [44,47]. According to crystallographic data, each AnxA5 monomer can bind up to ten Ca^2+^ ions on the membrane-binding side, especially in the presence of negatively charged lipids such as phosphatidylserine (PS) [31].

The annexins are the most abundant family of proteins found present in the MV membrane, and the ability of AnxA5 in transporting Ca^2+^ ions from a more to a less concentrated microenvironment [26,28,29,42] may be directly related to the initiation of hypertrophic cartilage matrix mineralization [28]. However, the mechanisms regulating AnxA5-mediated Ca^2+^ influx into proteoliposomes or MVs are still not well understood.

Kirsch et al. [28] showed by means of fluorescence the Ca^2+^ influx into fura-2-loaded liposomes in the presence of AnxA5. This protein is able to modulate Ca^2+^ permeability into liposomes and is not simply due to leakiness of the membrane caused by the presence of protein, once it was shown that the Ca^2+^ influx was inhibited by the presence of antibodies to AnxA5 [28]. Years later, they showed the Ca^2+^ influx ability promoted by AnxA5, AnxA6, and AnxA2 into native MVs isolated from chondrocytes [42]. Bolean et al. [26] also were able to demonstrate the ^45^Ca^2+^ influx into 9:1 DPPC:DPPS (molar ratio) liposomes harboring AnxA5. AnxA5-proteoliposomes had the highest affinity for collagens, especially for type II, supportive of a role for AnxA5 during chondrocyte mineralization in joint cartilage [27,44].

AFM enables morphologic analysis of proteoliposomes at the nanoscale without the use of fluorescent dyes, just as it is widely used to image membrane structures by means of fluorescence microscopy. We need a high-resolution imaging technique that works on soft nanosystems with a minimal degree of perturbation of the system, and AFM brings this advantage when compared with other techniques. AFM enables recording of phase images by monitoring the shift between the phase angle of the cantilever oscillations relative to that of the drive signal. Since proteins and lipids have different physical properties, the presence of proteins within lipid membranes can be detected as surface regions with distinct phase angle shifts in AFM phase images, thus enabling the mapping of single proteins as well as protein aggregates on lipid membranes. Thus, darker spots on the surface of the proteoliposomes are domains with lower viscoelasticity when compared with the entire vesicle surface. This suggests that these darker dots are regions with different chemical composition induced by the AnxA5 protein presence in the lipid bilayer of proteoliposomes. Therefore, AFM data suggest the presence of AnxA5 in the proteoliposome, besides showing direct images of the liposomes and proteoliposomes as intact vesicles [44].

It was expected that after AnxA5 incorporation into the bilayer lipid of liposomes, thus forming proteoliposomes, we would obtain proteoliposomes vesicles with bigger diameters when compared with the same vesicles without protein inserted (liposomes) [26]. This behavior is also observed when other proteins, for example alkaline phosphatase, are anchored to the liposome vesicle through the glycosylphosphatidylinositol anchor [26,44].

Under the conditions of the measurements carried out using the BNCD microelectrodes, the molar ratio of Ca^2+^/AnxA5 was 50. Since it has been reported that the AnxA5 monomer binds up to 10 Ca^2+^ ions [31], this experimental condition guaranteed excess of Ca^2+^ in the solution and confirmed that the reduction in the potential was not only due to the binding of Ca^2+^ to AnxA5, but mainly to the transfer of ions to the lumen of the proteoliposomes, showing the transport function of AnxA5 when incorporated in the vesicles’ membrane bilayer.

The lipid DPPS plays a very important role both for improving AnxA5 incorporation into liposomes and for favoring binding of proteoliposomes to collagens [27]. Besides that, negative charges can influence the Ca^2+^ transport by AnxA5 into DPPS:DPPC:Chol (2:1:1) liposomes, raising the levels of mineral propagation in odontoblast pre-differentiated cells, leading to the odontogenic differentiation [48]. Therefore, AnxA5 and PS work synergistically both in the Ca^2+^ transport to the inside of vesicles and targeting them to collagen fibers in the extracellular matrix.

Results described by Fezoua-Boubegtiten et al. [49] in PS monolayers using polarization modulation infrared reflection absorption spectroscopy (PM IRRAS), quartz crystal microbalance (QCM) and simulation by molecular dynamics (MD) techniques, showed that both Ca^2+^ and Mg^2+^ ions bind to AnxA5, but only Ca^2+^ is able to support an orientation in the α-helix structures of the protein. While Ca^2+^ ions are fully accessible to water on the AnxA5 surface, some of the Mg^2+^ ions are embedded in the protein structure. Indeed, by means of BNCD microelectrodes it was shown here that AnxA5 is able to transport Ca^2+^ in a selective way when compared with Mg^2+^.

Based on the results obtained, it is possible to re-establish a correlation with the physiological condition during mineralization. Thus, during bone formation, osteoblasts or chondrocytes produce and mineralize the extracellular matrix (ECM). The formation of apatite is initiated in the lumen of MVs and has been described to depend on the presence of a nucleator, the nucleational core (NC), in the lumen of MVs. The components of the NC, and amorphous calcium phosphate (ACP) in association with PS, result in calcium–phosphate–lipid complexes (PS-CPLXs). Further accumulation of Ca^2+^ and inorganic phosphate (Pi) in MVs can lead to the growth of apatite crystals, which then propagate on the ECM and continue to grow onto collagenous fibers, forming bone tissues. More than any other protein so far studied, AnxA5 accelerates the nucleational activity of the acidic phospholipid-Ca-Pi complexes present in the NC, which triggers de novo calcium phosphate mineral formation in MVs. AnxA5 interacts with Ca^2+^ and the PS polar head groups present in the MV lipid membrane, and domains are created that apparently exclude Mg^2+^. Thus, in the presence of PS, AnxA5 shows no affinity for Mg^2+^, in strong contrast to its high affinity for the Ca^2+^ ion [50,51]. Thereby, if AnxA5 and PS micelles are co-incubated with low levels of Ca^2+^, they quickly form complexes that co-sediment with the PS liposomes. On the other hand, if Ca^2+^ is replaced for Mg^2+^ even at much higher levels, AnxA5 does not bind Mg^2+^ or co-sediment with the PS liposomes. This behavior is observed because the coordination chemistry of Mg^2+^ is very different from that of Ca^2+^ [52,53,54]. Whereas Ca^2+^ can form flexible bond lengths and angles with as many as seven different ligands, Mg^2+^ coordinates with only six ligands in a highly rigid, precise geometry [55,56]. Thus, Mg^2+^ displays a rigid coordination geometry, which blocks the nucleational activity of PS-CPLX, and consequently prevents ACP from forming hydroxyapatite or octacalcium phosphate crystals.

## 5. Conclusions

To date, there is no description of the possible ability of Mg^2+^ ions to be transported by AnxA5, and questions related to the role of AnxA5 in the transport of Ca^2+^ ions remain. Therefore, this paper describes the high sensitivity of BNCD microelectrodes to track small changes in Ca^2+^ concentration due to binding to proteoliposomes, which could not be investigated using common Ca^2+^ selective electrodes. The authors are aware that the reported results are essentially of a qualitative nature; therefore, future work will focus on providing more precise quantitative assessments. However, the results shown here support, with adequate evidence, the hypothesis of selective Ca^2+^ transport within the proteoliposomes and the possible function of AnxA5 mediating the mineralization initiation process that occurs in MVs.

## Figures and Tables

**Figure 1 biosensors-12-00525-f001:**
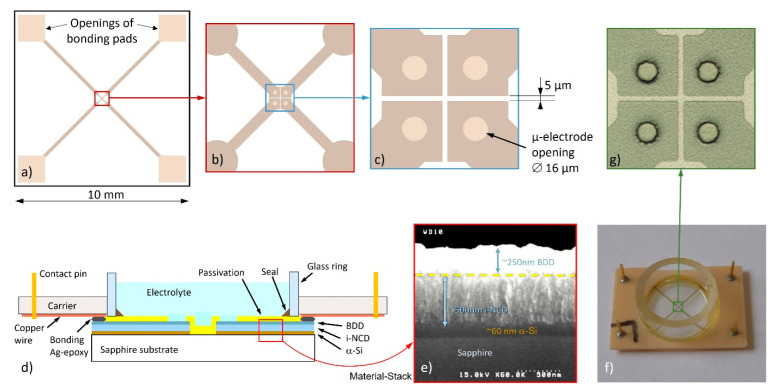
The microelectrodes chip. (**a**) Layout of the 10 × 10 mm device; (**b**,**c**) progressive magnifications of the core region with the microelectrodes; (**d**) scheme of device cross-section showing the various layers and features; (**e**) SEM-picture showing the cross-section of the material stack before the deposition of the passivation layer; (**f**) assembled chip with the 200 µL sensing chamber; (**g**) micrograph showing the same region depicted in (**c**) of the actual chip.

**Figure 2 biosensors-12-00525-f002:**
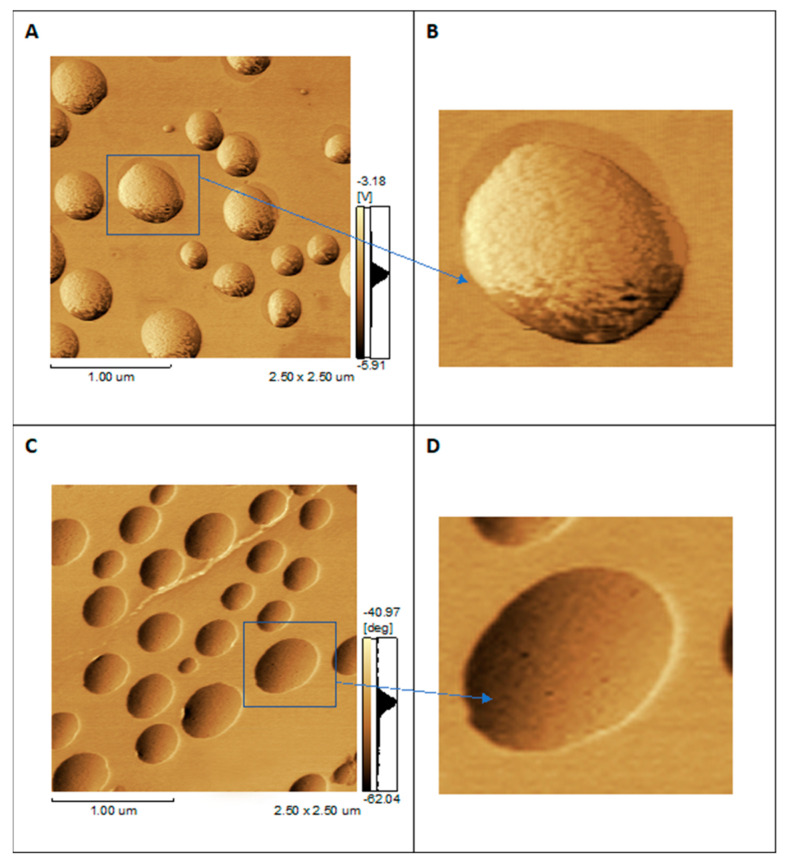
AFM phase images of 9:1 DPPC:DPPS (molar ratio) vesicles (0.6 mg mL^−1^) of (**A**) liposomes (2.5 × 2.5 μm scales); (**B**) zoomed detail of only one vesicle; (**C**) proteoliposomes harboring AnxA5 (1:100 protein:lipid, molar ratio) (2.5 × 2.5 μm scales); (**D**) zoomed detail of only one vesicle.

**Figure 3 biosensors-12-00525-f003:**
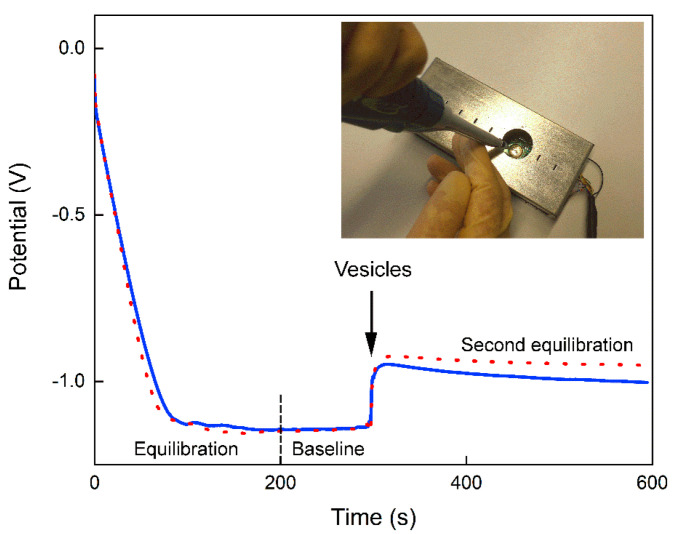
Potential (V) versus time (s) curves recorded using the BNCD microelectrode in the presence of deionized water. A volume of 10 µL of liposomes (0.6 mg/mL of total lipid) (dot line) or proteoliposomes (0.6 mg/mL of total lipid containing 14 µg/mL of AnxA5) (continuous line) were added to the microelectrode compartment after 300 s. After initial equilibration a nearly stable baseline was reached. Since this equilibration is not relevant for the signal analysis, the following figures focus on the time range from 200 to 600 s. The potential steps observed after dispensing the vesicles are due, in this case, to the Tris-HCl buffered vesicle solutions, since no other ions are present. Such steps drop down by a few tens of millivolts over the next 300 s, indicating a second equilibration transient. The inset shows the device inserted in the readout electronics inside the shield box and the dispensing of the vesicle sample by manual pipetting.

**Figure 4 biosensors-12-00525-f004:**
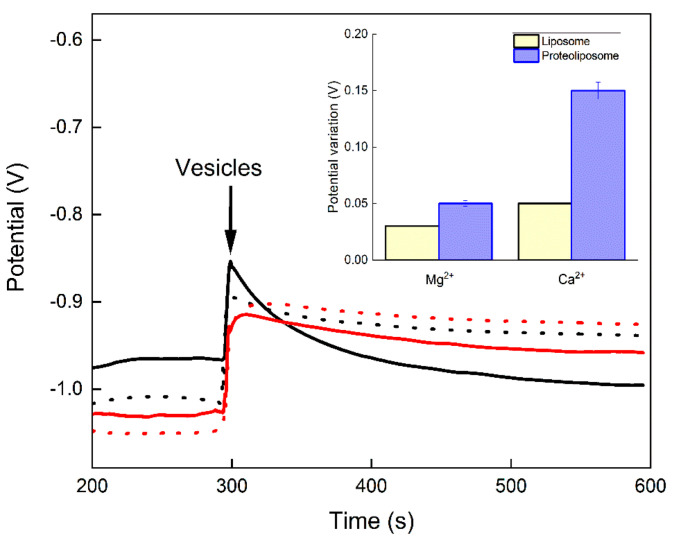
Potential (V) versus time (s) curve recorded with the BNCD microelectrode in the presence of either Mg^2+^ or Ca^2+^ ions in deionized water. After the first equilibration, 10 µL of liposomes (0.6 mg/mL of total lipid) (dot lines) or proteoliposomes (0.6 mg/mL of total lipid containing 14 µg/mL of AnxA5) (continuous lines) were added at 300 s to the microelectrode compartment. Black lines correspond to the potential recordings in the presence of 1 mM Ca^2+^ ions and red lines to potential recordings in the presence of 1 mM Mg^2+^ ions. Inset: Absolute values of the potential changes observed from 300 to 600 s, i.e., over the long-term transient after the addition of vesicles.

**Figure 5 biosensors-12-00525-f005:**
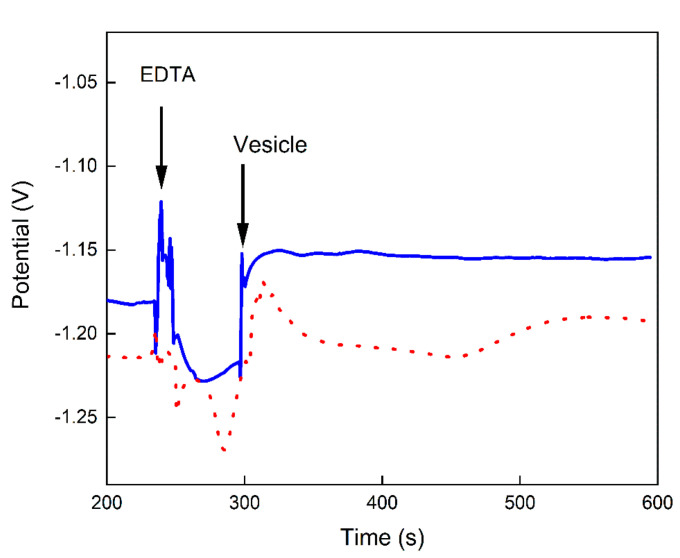
Potential (V) versus time (s) recorded with the BNCD microelectrode in media containing 1 mM Ca^2+^ ions in 16 mM Tris-buffer, pH 7.5. After equilibration, 4 µL of 2.5 mM EDTA were added at 240 s to the microelectrode followed by the addition of 10 µL of liposomes (0.6 mg/mL of total lipid) (dot line) or proteoliposomes (0.6 mg/mL of total lipid containing 14 µg/mL of AnxA5) (continuous line) at 300 s. The chelator prevents internalization of Ca^2+^ ions by the proteoliposomes and the potential fluctuates in a small range of just a few tens of mV. Due to the presence of the Tris-buffer from the beginning, the potential steps are much smaller than the ones shown in the previous figures. The signal fluctuations are artifacts due to vibrations and turbulences generated by the manual pipetting technique. Despite of such fluctuations, both recordings end at a level of ~30 mV above the baseline, which suggests the absence of ion transport due to the chelating action of EDTA.

**Figure 6 biosensors-12-00525-f006:**
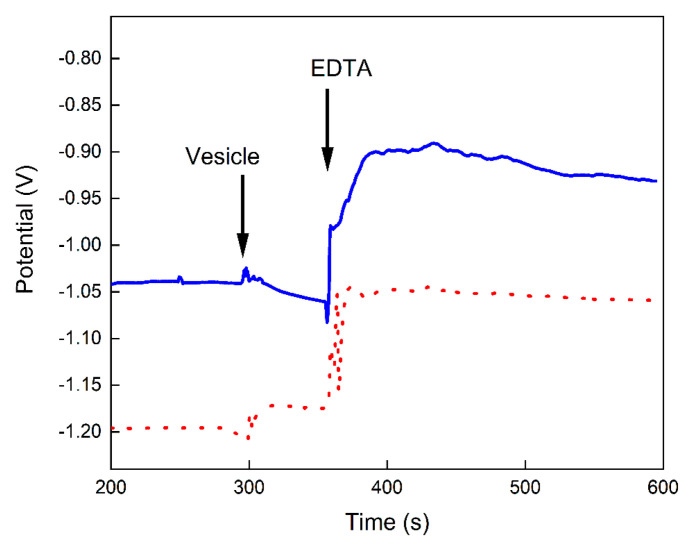
Potential (V) versus time (s) curve recorded using the BNCD microelectrode in media containing 1 mM of Ca^2+^ ions in Tris 16 mM, pH 7.5. After equilibration, at 300 s, 10 µL of liposomes (0.6 mg/mL of total lipid) (dotted line) and proteoliposomes (0.6 mg/mL of total lipid containing 14 µg/mL of AnxA5) (continuous line) were added, followed by the addition of 4 µL of 2.5 mM EDTA at 360 s. A potential drop of ~30 mV is clearly visible from 300 to 360 s in the recording with proteoliposomes.

**Table 1 biosensors-12-00525-t001:** Summary of all measurement variations and the respective results. Recordings 1 to 4 and 17 are negative controls, i.e., without liposomes/proteoliposomes. All measurements with Tris-buffer in the initial solution present smaller potential variations due to the buffering effect.

#Recording	Aim	Initial Solution	1st Dispensing	2nd Dispensing	Results
1	Test capacitive effects of volume changes	1 mM Ca^2+^ in water	1 mM Ca^2+^ in water	-	Stable potential
2		1 mM Mg^2+^ in water	1 mM Mg^2+^ in water	-	
3	Test potential change by buffer dispensing	1 mM Ca^2+^ in water	Tris (16 mM pH 7.5)	-	~150 mV step with second equilibration
4		1 mM Mg^2+^ in water	Tris(16 mM pH 7.5)	-	
5	Test liposome response to Mg^2+^	1 mM Mg^2+^ in water	Tris + Lipo	-	~150 mV step with second equilibration
6	Test proteoliposome response to Mg^2+^	1 mM Mg^2+^ in water	Tris + Proteo	-	~130 mV step with small drop vs. time
7	Test liposome response to Ca^2+^	1 mM Ca^2+^ in water	Tris + Lipo	-	~140 mV step with second equilibration
8	Test proteoliposome response to Ca^2+^	1 mM Ca^2+^ in water	Tris + Proteo	-	~120 mV step, drops to baseline
9	Test liposome response to Mg^2+^ in buffer	1 mM Mg^2+^ in Tris	Tris + Lipo	-	~40 mV step with second equilibration
10	Test proteoliposome response to Mg^2+^ in buffer	1 mM Mg^2+^ in Tris	Tris + Proteo	-	~30 mV step with second equilibration
11	Test liposome response to Ca^2+^ in buffer	1 mM Ca^2+^ in Tris	Tris + Lipo	-	~40 mV step with second equilibration
12	Test proteoliposome response to Ca^2+^ in buffer	1 mM Ca^2+^ in Tris	Tris + Proteo	-	~30 mV potential step, drops to baseline
13	Test liposome response to Na^+^	1 mM Na^+^ in water	Tris + Lipo	-	~220 mV step with second equilibration
14	Test proteoliposome response to Na^+^	1 mM Na^+^ in water	Tris + Proteo	-	~200 mV step with second equilibration
15	Test liposome response to Na^+^ in buffer	1 mM Na^+^ in Tris	Tris + Lipo	-	~20 mV potential step
16	Test proteoliposome response to Na^+^ in buffer	1 mM Na^+^ in Tris	Tris + Proteo	-	~10 mV potential step
17	Test potential change by chelator dispensing	1 mM Ca^2+^ in Tris	EDTA		~20 mV negative step ~stable over time
18	Test chelator effect on liposome response to Ca^2+^	1 mM Ca^2+^ in Tris	Tris + Lipo	EDTA	(1) ~20 mV steady step (2) ~130 mV steady step
19	Test chelator effect on proteoliposome response to Ca^2+^	1 mM Ca^2+^ in Tris	Tris + Proteo	EDTA	(1) ~30 mV dropping slope (2) ~150 mV step with second equilibration
20	Test chelator effect on liposome response to Ca^2+^	1 mM Ca^2+^ in Tris	EDTA	Tris + Lipo	(1) ~40 mV negative step (2) step ends ~30 mV above baseline
21	Test chelator effect on proteoliposome response to Ca^2+^	1 mM Ca^2+^ in Tris	EDTA	Tris + Proteo	(1) ~40 mV negative step (2) step ends ~30 mV above baseline
22	Test chelator effect on proteoliposome response to Ca^2+^	1 mM Ca^2+^ in Tris + EDTA	Tris + Proteo		~30 mV steady step

## Data Availability

The data presented in this study are available on request from the corresponding authors.

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
