# Peer review of "Ultrasensitive Diamond Microelectrode Application in the Detection of Ca2+ Transport by AnnexinA5-Containing Nanostructured Liposomes"

_biosensors, 2022, doi:10.3390/bios12070525_

Round 1

Reviewer 1 Report

The manuscript Ultrasensitive diamond microelectrode application in the detection of Ca2+  transport by AnnexinA5-containing nanostructured
liposomes revisits earlier work (ref 15 where AnxA5 was demonstrated able to mediate Ca2+-influx into the DPPC and DPPC:DPPS 10%-vesicles at physiological Ca2+ concentrations (∼2 mM) or ref 3 2010 Nano-
crystalline Diamond Microelectrode Arrays Fabricated on Sapphire Technology) to "support the hypothesis of a selective Ca2+
 transport within AnxA5 proteoliposomes and the possible
function of  mediating the mineralization initiation process that occurs in MVs."- according to conclusions.

The manuscript is not suitable for publication for the following reasons:

-the degree of originality and novelty is reduced.

-It contains copied or irrelevant data: figure 1 C is copied from ref 33! from 2017. It is not clear what is the relevance of AFM images on mica and on crosslinked liposomes when the microelectrode data are achieved on boron-dopped crystalline diamond. These data are missing.

-it contains "uncurated data": Fig. 1 shows phase images of liposomes and proteoliposomes harboring AnxA5 with different scales of the AFM images (V) versus degrees. All evolutions in figures 2-5 are missing statistical analysis (to provide mean evolutions and standard deviations).

-It contains vague explanations:

*section 2.6 concerning the microelectrodes contains the fabrication steps that were previously reported but it does not provide actual data on: thickness and quality of the layers in the microelectrodes structure, no detail on the actual geometry and time stability upon cleaning steps and various solutions injections.

* figure 1 explanations in lines 167-170 are missing quantitative terms instead of "darker domains" that have no scientific meaning.

*caption figure 2 states 0.6 mg/mL of total lipid containing 14 μg/mL of AnxA5 and (lines 274-275) Under the conditions of the measurements carried out using the BNCD microelectrodes, the molar ratio of Ca2+/AnxA5 was 50. How is this information related to the experimental conditions line 96 "about 1:100 protein lipid molar ratio" and 10 ul injection in a 1mM Ca2+ 200ul volume is difficult to process without the details on the molecular weight and the actual volumes used.

-It contains ill-designed experimental procedures:

figure 2 contains data that have no error bars AND are related to distilled water as the buffer, whereas liposomes are achieved in Tris HCl buffer and injected at around 1mM concentration. Ph changes of Tris buffer due to dillution, osmolarity changes of the proteoliposomes and different electrodes behaviour are combined detrimental to concept clarity in this experimental procedure. Statistically meaningful data for Tris HCl buffer with and without Ca and vesicles are needed together with blank injections of stock buffers with or without Ca instead of statements like "Similar behavior was found with samples dispersed in either deionized water or 16 mM Tris buffer, pH 7.5, even in the presence of 2.5 mM NaCl (data not shown)".

*Blank injections data and details on the mixing are all the more needed since it is not clear how the homogenization upon injection of 10ul sample is achieved in the 200 ul total volume. The position of injection in respect to the active electrode is not evident in the analysis.

-Figure 3 data are also related to distilled water as buffer. Intriguingly, whereas the data do not contain error bars, there is both a difference in the stabilisation levels prior to injections and a different dynamics after vesicle injections. The meaning of the inset "Potential variation" is thus not clear: variation between what levels, at which time points? and it is also difficult to relate it with the main figure for instance for Mg in distilled water or when comparing Mg and proteoliposomes data with Ca and liposomes.

Figures 4 and 5, while in Tris buffer as they should, lack statistical control and are difficult to be integrated with a coherent mechanistic understanding: stabilisations prior to injections have radically different values of the potential (if one compares continuous line figures 4 and 5) and dynamics upon injection (if one compares continuous line figure 5 and black continuous line figure 3).

The nonlinear evolutions visible in figures 4 and 5 and the proposed effect of Ca and EDTA lines 220-226 are seemingly incongruent:

-the "faster reduction" of the potential upon addition of EDTA (line 220) is not clear.

-How is the statement in lines 222-224 "in this way the changes in the potential after the addition of proteoliposomes to the microelectrode in
presence of EDTA is similar to the changes observed in the absence of ions (Fig. 4)" cover the behaviour for plane liposomes? the dotted line in figure 4.

-What process is related to the evolution upon EDTA addition subsequent to liposomes injection (figure 5).

The discussion section should have integrated details on the results presented. As it is now it is marginally related to the results or the Table. Lines 275-280 require a thorough address of the results.

-the actual evolutions related to Table summary, that should have been included as supplementary materials, should provide quantitative details to replace the "small", "positive", "negative" statements in Table.

Reviewer 2 Report

This work presented the interesting application of high sensitivity boron-doped nanocrystalline diamond microelectrodes, for tracking small changes in Ca2+ concentration due to the binding to Annexin-A5 inserted into the lipid bilayer of liposomes (proteoliposomes). The results could confirm the hypothesis of a selective Ca2+ transport within the proteoliposomes through the function of AnxA5 protein. This work is well written presenting an innovative way to study Ca2+ transport potentially in bioapplications. So, I suggest to accept this work after minor revision.

1. In part 3.1 Characterization of liposomes and proteoliposomes, the standard deviation of mean diameter and ζ-potential must present with the sign “±” not “+”

2. The authors should comment on the bigger size of proteoliposomes compared with the size of liposomes in the manuscript.

3. The AFM phase images of liposomes and proteoliposomes (Figure 1A and 1C) should be showed in the same scale.

4. It is quite difficult to confirm the formation of surface protrusions by using only AFM technique. The authors should further confirm the morphology of proteoliposomes with other techniques such as SEM or TEM.

5. The authors should clearly highlight why the selective transport of Ca2+ over Mg2+ is important for this study    

Reviewer 3 Report

The current manuscript entitled “Ultrasensitive diamond microelectrode application in the detection of Ca2+ transport by AnnexinA5-containing nanostructured liposomes” by “Pasquarelli et al” deliberated on the innovative application of high sensitivity Boron-doped nanocrystalline diamond microelectrodes, for tracking small changes in Ca2+ concentration due to the binding to Annexin-A5 inserted into the lipid bilayer of liposomes which could not be assessed using common Ca2+ selective electrodes. The data provided evidenced the selective Ca2+ transport into the proteoliposomes and support the possible function of AnxA5 as a hydrophilic pore once incorporated into lipid membrane, mediating the mineralization initiation process occurring in matrix vesicles. The work seems good and interesting. The manuscript can be accepted after addressing the following comments.

1.      In the introduction, authors should discuss on the existing articles related to the work.

2.      On line number 41, please provide the list of applications of BCND microelectrodes.

3.      Please provide some more discussion of the work, seems discussion part is less.

Round 2

Reviewer 1 Report

N/A